# Navigating Beyond Instructions: Vision-and-Language Navigation in Obstructed Environments

## ABSTRACT

Real-world navigation often involves dealing with unexpected obstructions such as closed doors, moved objects, and unpredictable entities. However, mainstream Vision-and-Language Navigation (VLN) tasks typically assume instructions perfectly align with the fixed and predefined navigation graphs without any obstructions. This assumption overlooks potential discrepancies in actual navigation graphs and given instructions, which can cause major failures for both indoor and outdoor agents. To address this issue, we integrate diverse obstructions into the R2R dataset by modifying both the navigation graphs and visual observations, introducing an innovative dataset and task, R2R with UNexpected Obstructions (R2R-UNO). R2R-UNO contains various types and numbers of path obstructions to generate instruction-reality mismatches for VLN research. Experiments on R2R-UNO reveal that state-of-the-art VLN methods inevitably encounter significant challenges when facing such mismatches, indicating that they rigidly follow instructions rather than navigate adaptively. Therefore, we propose a novel method called ObVLN (Obstructed VLN), which includes a curriculum training strategy and virtual graph construction to help agents effectively adapt to obstructed environments. Empirical results show that ObVLN not only maintains robust performance in unobstructed scenarios but also achieves a substantial performance advantage with unexpected obstructions. The source code is available at https://anonymous.4open.science/r/ObstructedVLN-D579.

## CCS CONCEPTS

• **Computing methodologies** → **Computer vision tasks**; • **Information systems** → **Multimedia information systems**.

## KEYWORDS

vision-and-language navigation, embodied agents, object insertion

## 1 INTRODUCTION

Vision-and-Language Navigation (VLN) [5] requires agents to follow natural language instructions to reach a specified destination. Recently, there has been a growing interest in this area due to its great potential for real-world applications such as household robots, especially with the booming of Large Language Models (LLMs) [9, 63]. However, current VLN tasks are limited by several unrealistic assumptions, making their application remain in simulators with few real-world robotic deployments [18, 70].

One significant constraint is what we call the "**perfect instruction assumption**", which implies that instructions are always perfectly aligned with the environment, neglecting real-time dynamics like unexpected obstructions. Agents trained under this assumption excel in following instructions but lack adaptability for actual navigation where discrepancies between instructions and reality are commonly seen. For example, as shown in Fig. 1, the human instructs the agent to *"Walk straight down the hall"* based on

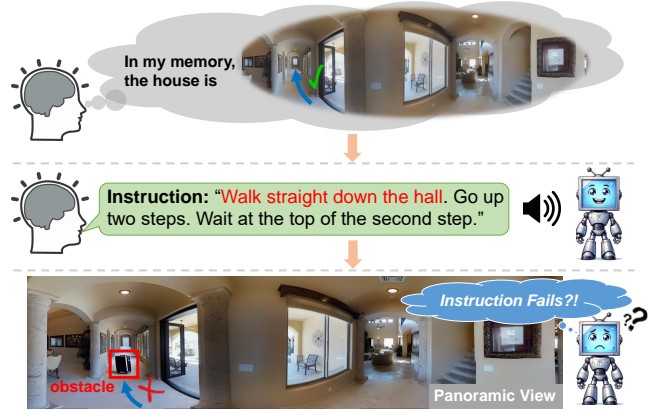

**Figure 1: Discrepancy between instructions and reality in real-world navigation. The instructions from humans are based on prior memory and often can not align with real-time environments. Current VLN environments overlook this mismatch, potentially causing navigation failure.**

prior knowledge of the house. However, real-world environments have changed, with unforeseen obstacles like a suitcase blocking the hallway. While humans can quickly adapt and find detours, current VLN agents struggle with these instruction-reality mismatches, often leading to navigation failures.

To address this issue, it is essential to introduce these discrepancies into VLN. While various factors can lead to such discrepancies, this work focuses on one of the most representative and prevalent causes: obstructions. We propose to integrate obstructions into existing discrete VLN environments to block the path described by the instruction, resulting in an instruction-reality mismatch. Notably, although both approaches involve obstructions, our work differs from previous work on obstacle avoidance [3, 71] in terms of problem setup, navigation focus, and potential solutions (See Sec. 2), leading us to choose the discrete setting.

Therefore, we make various modifications to the navigation graphs and visual observations of the R2R dataset [5], proposing the R2R with UNexpected Obstructions (R2R-UNO) dataset as the first VLN task to emphasize the instruction-reality mismatches. At the graph level, we selectively block edges whose removal does not impact the overall graph connectivity in existing paths, ensuring agents can still reach the destination. To keep visual coherence with graphs, we design an object insertion module that leverages text-to-image inpainting techniques [58] to integrate diverse objects into different scenes seamlessly. Due to the instability of inpainting results, a filtering module is employed to select the high-quality ones from multiple candidates to ensure obstruction generation. Our study using R2R-UNO reveals that state-of-the-art VLN agents [13, 68] perform poorly in obstructed environments, limiting their practical applications.

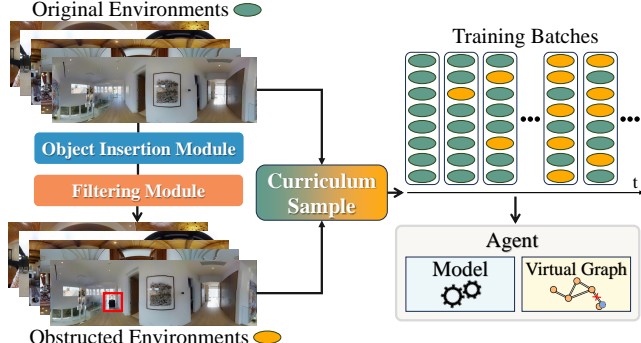

**Figure 2: The overall framework of our method. We first generate obstructed environments based on existing datasets, and then train agents with our proposed curriculum strategy and graph construction mechanism on both data types.**

Although we expect agents to perform well with mismatches, it is crucial to maintain their performance in original environments simultaneously, which represent most situations. However, directly training with these two types of data poses challenges for agents to learn beyond following instructions and to differentiate between mismatched and aligned scenarios. Empirical results show that discrepancies between these two environments can lead to imbalanced optimization, favoring one type. To address this, we develop the Obstructed VLN (ObVLN) method, including a curriculum training strategy [7] to organize the training and a novel graph construction mechanism to introduce virtual nodes for blocked edges to facilitate efficient exploration. The overall framework including data generation and agent training is presented in Fig. 2.

We conduct comprehensive experiments on R2R, REVERIE [52], and R2R-UNO datasets to demonstrate the significance of incorporating instruction-reality mismatches in VLN. Established methods, such as DUET [12], struggle in obstructed settings, with a significant 30% drop in Success Rate (SR). The proposed object insertion and filtering modules provide crucial visual feedback aligned with navigation graph changes, proved by the improved performance of agents using inpainting images. By employing ObVLN, agents not only maintain robust performance in original scenarios but also effectively adapt to mismatches, achieving an impressive 67% SR, marking a significant advancement.

We summarize our contributions in this paper as follows:

- We address the underexplored issue of instruction-reality mismatches in VLN by proposing R2R-UNO, the first VLN dataset that includes such mismatches through graph changes and diverse obstruction generation, offering a unique challenge that reflects real-world navigation complexities.
- We highlight the lack of adaptability in current VLN methods for obstructed environments and propose ObVLN as a solution, which employs curriculum learning and virtual graph construction to enhance agent adaptability.
- Through extensive experiments, we prove the significance of introducing R2R-UNO in VLN research and show that Ob-VLN performs well in both original and obstructed environments, achieving a significant 23% SR increase in R2R-UNO.

## 2 RELATED WORK

*Vision-and-Language Navigation.* In VLN, agents need to navigate within simulated environments like Matterport3D [10] following natural language instructions. Numerous methods [19, 47, 53, 61, 74] and datasets [33, 43, 52, 62, 73] have been developed to address various challenges in this domain. Some works address the data scarcity problem by introducing extra sources, including synthesized instructions [66, 67], additional environments [29, 41], and predicted scenes [31, 37]. Recently, ScaleVLN [68] synthesized enormous high-quality instruction-trajectory pairs for HM3D [55] and Gibson [69] environments to make agent performance approach human results. Other works [24, 40, 46] apply diverse network structures such as LSTM [23], Transformer [64], and Graph Neural Networks [21] to enhance cross-model alignment and decision-making ability. For example, HAMT [11] encodes the full history information through transformers and integrates it with instructions and observations for better action prediction. DUET [12] further maintains a topological map to enable agents aware of global visual representations to make global decisions instead of adjacent viewpoints. We adopt HAMT and DUET as the evaluation models due to their superior performance and difference in whether map-based.

*Environment Changes in VLN.* Many works [16, 34, 51] have proposed to modify VLN environments for different purposes, which can generally be categorized based on the changing level. One category involves changing the visual observations to augment data for improving agent generalization. For example, Li et al. [17] utilized image captioning to obtain language descriptions of panoramas and employed generative models to produce novel views based on these descriptions. Conversely, another category focuses on graph-level adjustments rather than the visual level to go beyond the established navigation graphs by mixing up different graphs [42] or substituting a fixed viewpoint with proximate locations [31]. Notably, VLN-CE [32] abandons the graph-based navigation paradigm and allows agents to traverse continuous environments freely to enhance task realism. Differently, our task makes changes at both visual and graph levels to block edges in the graph and generate obstacles in the views to generate instruction-reality mismatches.

*Obstacle Avoidance.* Obstacle avoidance has been a long-standing research challenge in visual navigation [1, 8, 49, 59]. In VLN, ETP-Nav [3] applies an obstacle-avoiding controller with a trial-and-error heuristic to explicitly escape from deadlocks. SafeVLN [71] employs a LiDAR-based waypoint predictor and a re-selection strategy to avoid non-navigable waypoints and obstacles. However, our work diverges from these works under the continual setting [32] in three aspects, leading us to choose the discrete setting. Firstly, we target a broader range of situations with instruction-reality mismatches, including avoidable obstacles, closed doors, rearranged furniture, instruction errors, etc. These scenarios can be effectively captured by graph changes but are hard to model as continuous signals. Secondly, while obstruction avoidance focuses on assessing the properties of obstructions to avoid them, our work emphasizes changes in navigation graphs that render the instructions temporarily invalid, ignoring the nature of the obstructions. The discrete setting decouples path planning from technicalities, such as the

shape and size of obstacles, thereby focusing on high-level adaptability when instructions fail and enhancing its generality. Finally, addressing instruction-reality mismatches involves more than just navigating around obstructions. It requires a complex strategy to find detours and the ability to navigate without instruction guidance, posing a significant adaptability challenge which is too hard for current continuous agents.

*Object Insertion.* Inserting novel objects into target images has always been a research challenge in Computer Vision. Early works [28, 30, 35] employ the cut-and-paste strategy to merge two pictures, which is straightforward but lacks photorealism. With the development of neural networks, object insertion can be achieved in various ways and higher quality, including image synthesis [36], neural radiance fields editing [65], image generation [6], etc. In navigation, THDA [48] inserts 3D scans of household objects into random locations as navigation goals to augment the training data for ObjectGoal Navigation [4]. In VLN, Envedit [16] leverages semantic image synthesis [50] to create objects based on modified environment semantics as augmented environments. In this work, we uniquely take advantage of the text-to-image inpainting models [58] to seamlessly embed desired objects within specific view locations without influencing visual surroundings.

## 3 OBSTRUCTED ENVIRONMENTS

In this section, we first introduce the problem formulation of VLN on vanilla and obstructed environments, and then describe our proposed R2R-UNO dataset for introducing instruction-reality mismatches into VLN.

### 3.1 Problem Setup

In VLN, the agent must follow a natural language instruction $\hat{x} = (x_1, x_2, ..., x_L)$ with $L$ words to navigate a simulated environment. This environment is usually discrete with a predefined undirected navigation graph $\mathcal{G} = \{\mathcal{V}, \mathcal{E}\}$ with navigable nodes $\mathcal{V}$ and connectivity edges $\mathcal{E}$. At each timestep $t$, the agent perceives a panoramic representation, comprised of $N = 36$ views $O_t = \{o_t^i\}_{i=1}^N$ and an orientation feature that encodes heading $\theta$ and elevation $\phi$ information of its current node $v_t$. It then determines an action $a_t$ to transition to one of the neighboring nodes $\mathcal{N}(v_t)$ by selecting the view that aligns best with the target node from the current panorama $O_t$. An additional "STOP" action is available to conclude the navigation process. For each instruction $\hat{x}$, there exists a corresponding ground truth path $P = < v_1, e_1, v_2, e_2, \cdots, v_n >$ representing the intended trajectory with $n$ nodes for the agent. However, previous work presumes perfect alignment between instruction and reality, assuming that all edges $< e_1, e_2, \cdots, e_{n-1} >$ are consistently accessible. This assumption overlooks the dynamics of real-world navigation, where the graph $\mathcal{G}$ may have changed. Among various reasons for graph changes, we focus on the most representative one: obstructions. In our obstructed environments, some specific edges $E_s \subseteq P$ might be obstructed, making parts of the instruction inapplicable to the current scenario. This obstruction also leads to corresponding visual changes in the panoramic views $O_t$ of nodes linked by these edges. When meeting obstructed edges $E_s$, the agent must find a substitute path $P'$ to reach the final destination $v_n$.

## Table 1: Statistics for the paths in R2R and R2R-UNO dataset.

| Dataset | Set | Num | Mean | Min | Max |
|---------|-----|-----|------|-----|-----|
| R2R | - | 5798 | 6.00 | 4 | 7 |
| R2R-UNO | Block-1 | 22982 | 8.15 | 5 | 15 |
| R2R-UNO | Block-2 | 37900 | 9.67 | 6 | 20 |
| R2R-UNO | Block-3 | 33599 | 11.13 | 7 | 25 |

### 3.2 R2R-UNO

In this section, we introduce the proposed R2R-UNO dataset in detail, including the graph changes and visual modifications.

*3.2.1 Graph Changes.* For a path $P = < v_1, e_1, v_2, e_2, \cdots, v_n >$ in R2R, we first identify all redundant edges within this path, denoted as $E_r \subseteq P$. A redundant edge $e_i = (v_i, v_{i+1}) \in E_r$ is defined by the property that its removal does not affect the overall connectivity of the graph, ensuring an alternate path $P'$ between $v_i$ and $v_{i+1}$. To account for scenarios where multiple edges are concurrently obstructed, we assess the collective redundancy of combinations of these redundant edges $E_r$ and categorize these combinations by the number of obstructed edges $x$ into Block-$x$ set. Since R2R paths have 4 to 7 nodes in length, we set $N_{max} = 3$ as the maximum number of obstructed edges, generating three sets Block-1, 2, 3, each containing individual training and evaluation splits.

With a Block-$x$ set, consider a combination of redundant edges $C = < e_{i_1}, e_{i_2}, \cdots, e_{i_x} >$. For each edge, we identify the shortest path $< P_1', P_2', \cdots, P_x' >$ between the corresponding nodes in the modified graph $\mathcal{G} = \{\mathcal{V}, \mathcal{E} - C\}$ to replace $C$ in the original path $P$, generating the path $\bar{P}$ as the new intended path for agents:

$$\bar{P} = \langle v_1, \ldots, v_{i_j}, P_j', v_{i_j+1}, \ldots, v_n \rangle, \quad \forall j \in 1, 2, \ldots, x \qquad (1)$$

We refer to $\bar{P}$ as the "*real path*" to be traversed by agents, in contrast to the "*instructional path*" $P$ based on the navigation instructions $\hat{x}$. Therefore, the instruction-reality mismatch is defined as $\bar{P} \neq P$, which demands agents actively seek alternate paths. To exclude excessively lengthy and impractical new routes, each Block-$x$ set possesses a "*real path*" length restriction as $L(\bar{P}) \leq 10 + 5 \cdot x$. As a result, 98.8% R2R paths are modified to be one or more new paths in R2R-UNO. As indicated in Tab. 1, the R2R-UNO sets contain a greater number and length of paths than R2R, with the average path length increasing with $x$, denoting greater navigation difficulty.

Notably, when constructing $\bar{P}$, we deliberately avoid shortening the path when $P'$ contains future viewpoints from $P$, even if this leads to the loop formation in $\bar{P}$. This design aligns with practical navigation scenarios since when deviating from the instruction, it is challenging for agents to align the current viewpoint with the described path due to the absence of contextual clues. Therefore, it is logical and efficient for agents to seek a detour around the obstruction before attempting to re-align with the instructions.

*3.2.2 Visual Changes.* To align with the graph changes, we introduce two novel modules to infuse various objects into panoramic views of nodes along redundant edges, as shown in Fig. 3.

The first one is the **object insertion module**, which employs a stable diffusion inpainting model [58] to approach the issue from an inpainting perspective. Consider a redundant edge $e$ linking nodes $v_a$ and $v_b$. We elaborate the process of modifying the panoramic

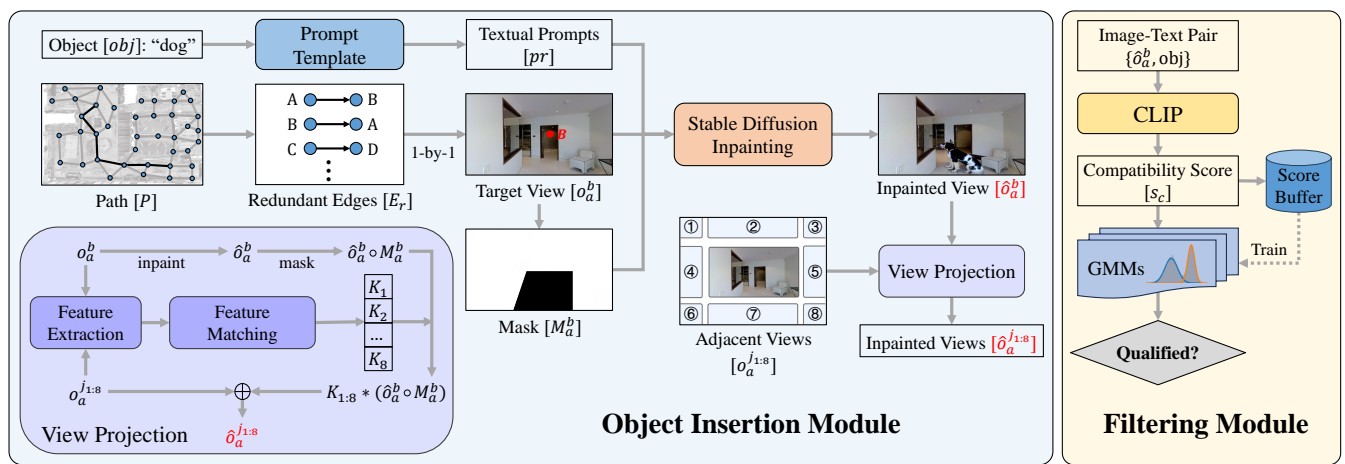

**Figure 3: The object insertion (left) and filtering module (right) in generating R2R-UNO. The red dot ● marks the position of node B in the view of node A; the ○ operator represents pixel-wise multiplication, while the ∗ symbol indicates pixel-wise matrix multiplication applied to image coordinates. The notation $j_{1:8}$ covers eight adjacent views ($j_1$ to $j_8$). The final images are highlighted in red. The dotted line from the score buffer illustrates the training process with all compatibility scores.**

view $O_a = \{o_a^i\}_{i=1}^{36}$ at node $v_a$ as follows, which is similarly applied to $O_b$. First, we localize the other node $v_b$ within $O_a$ to find the corresponding discrete view $o_a^b$ and calculate the pixel coordinates $(x_b, y_b)$ in $o_a^b$ as follows:

$$f = H/(2 \cdot \tan(\delta/2)) \qquad (2)$$

$$x_b = \tan(\theta_r) \cdot f + W/2 \qquad (3)$$

$$y_b = \tan(\phi_r) \cdot f + H/2 \qquad (4)$$

$W, H$ is the width and height of $o_a^b$, while $\theta_r$ and $\phi_r$ is the relative heading and elevation from $v_a$ to $v_b$, and $\delta$ is the vertical Field of View (FoV) to calculate the focal length $f$. A right trapezoid mask $M_a^b$ is then generated, expanding from the bottom edge to around the point $v_b$ with a specified width to cover potential areas of paths from $v_a$ to $v_b$. This is based on the observation that paths in these views either stretch from the bottom middle to the target point $(x_b, y_b)$ or form a vertical line from the bottom to the target. Finally, we combine an object name $obj$ with a predefined prompt template to formulate the final prompts $pr$, which, along with the view image $o_a^b$ and mask $M_a^b$, serve as the input to the inpainting model to generate the inpainted view $\hat{o}_a^b$:

$$\hat{o}_a^b = \text{Inpainting}(o_a^b, M_a^b, pr) \qquad (5)$$

While inpainting models can generate visually coherent and photo-realistic images, their instability is notable, with a certain probability of failing to incorporate the desired object. This instability often arises when the area outside the mask already contains elements similar to the intended object, such as a *"chair"* or *"table"*, which can mislead the inpainting model even with intricate prompts. Therefore, we propose a **filtering module** to improve the quality of inpainted images, which evaluates multiple generated candidates with different objects to filter out higher-quality ones. Specifically, to enrich the generated scenarios, we first carefully select ten object categories as obstructions based on their frequencies in both the Matterport3D dataset and real-world environments, including *chair, table, sofa, potted plant, basket, exercise equipment,*

*vacuum cleaner, suitcase, toy,* and *dog*. We then utilize the object insertion module to produce a set of novel views as candidates, each with a different kind of object. To reflect the inpainting quality, we employ the CLIP [54] model to assess the compatibility score $s_c$ of each view-object pair. Note that this assessment considers only the modified part of the view by combining the mask $M_a^b$:

$$s_c = \text{CLIP}(\hat{o}_a^b \circ M_a^b, obj) \qquad (6)$$

The ○ operator represents pixel-wise multiplication. These scores are aggregated to form a dataset including all redundant edges within R2R. Visual analyses of this dataset indicate that the score distribution for each object category often takes the shape of a bimodal Gaussian distribution, with one peak $\mathcal{N}(s_c|\mu_1, \sigma_1^2)$ for successful incorporation and another $\mathcal{N}(s_c|\mu_2, \sigma_2^2)$ for failure:

$$P(s_c) = \pi_1 \mathcal{N}(s_c|\mu_1, \sigma_1^2) + \pi_2 \mathcal{N}(s_c|\mu_2, \sigma_2^2) \qquad (7)$$

$\pi$ is the mixing coefficient, $\mathcal{N}(\cdot|\mu, \sigma^2)$ represents a Gaussian distribution with mean $\mu$ and variance $\sigma$. Based on this insight, we train a bimodal Gaussian Mixture Model (GMM) [57] for each category and use them to decide which images are qualified:

$$q = \begin{cases} 1 & \text{if } \mathcal{N}(s_c|\mu_1, \sigma_1^2) > \mathcal{N}(s_c|\mu_2, \sigma_2^2) \\ 0 & \text{otherwise} \end{cases} \qquad (8)$$

Finally, we randomly select one of those qualified candidates as the modified view, or, in the absence of suitable options, the choice is narrowed to the top three scorers. This module not only largely enhances the probability of incorporating visible obstructions but also ensures the selection of contextually fitting objects. We present the distributions of object categories and their compatibility scores in the appendix to prove the diversity of R2R-UNO.

All steps above only concern one discrete view $o_a^b$. The final step is propagating the updated view $\hat{o}_a^b$ to adjacent views $o_a^{j_{1:8}}$ containing overlapped areas with $o_a^b$ to maintain consistency across the panorama. We first calculate the transformation matrix $K_i$ for view $o_a^{j_i}$ based on SIFT features [45] matching. We then apply $K_i$ to

project the inpainted part of $\hat{o}_a^b$ onto the adjacent view and merge it with the original view to construct a coherent novel panorama:

$$K_i = \text{MATCH}(\text{SIFT}(o_a^b), \text{SIFT}(o_a^{j_i})) \qquad (9)$$

$$\hat{o}_a^{j_i} = o_a^{j_i} \cdot (1 - K_i * M_a^b) + K_i * \hat{o}_a^b \cdot (K_i * M_a^b) \qquad (10)$$

Here, $*$ denotes the projection operation by pixel-wise matrix multiplication applied to image coordinates.

It is important to note that we only perform 2D inpainting for nodes linked by redundant edges, which may lead to multi-view inconsistency. Although 3D object insertion can alleviate this problem, we stick to the 2D method for two reasons. Firstly, current 3D techniques [20, 56] mainly rely on limited pre-built object datasets [14, 39], which lack diversity and can cause visual inconsistencies in aspects like lighting and style. Other generative 3D methods [38, 60] struggle to generate high-resolution photo-realistic images and often produce noticeable artifacts. Secondly, the impact of multi-view inconsistency is minimal in our task, as instruction-reality mismatches are primarily defined by graph changes. Agents can only detect graph changes when moving to nodes alongside redundant edges, making visual modifications at other nodes irrelevant since agents should still follow instructions without detecting the mismatch. Moreover, various environment augmentation methods [16, 17] have proved that this inconsistency would not affect agent performance in real-world scenarios.

## 4 INSTRUCTION-REALITY MISMATCHES AND SOLUTION

In this section, we show that current VLN methods perform poorly when encountering instruction-reality mismatches and thus propose the ObVLN method to solve this problem.

### 4.1 Current VLN Methods Struggle in R2R-UNO

To assess the impact of instruction-reality mismatches, we conduct zero-shot evaluations on the R2R-UNO validation unseen splits using five advanced VLN methods, known for their excellent performance under the perfect instruction assumption: **1. RecBERT** [25] employs a recurrent BERT [15] model to preserve cross-modal information throughout navigation. **2. HAMT** [11] utilizes a transformer-based model to encode instructions, observations, and historical context. **3. DUET** [12] constructs a real-time topological map to enable global action decisions with a graph transformer. **4. GELA** [13] enhances the cross-modal alignment between visual landmarks and corresponding phrases with new annotations and pretraining objectives. **5. ScaleVLN** [68] integrates extensive high-quality environments and instruction-trajectory pairs from diverse datasets to enhance performance. Fig. 4 illustrates how their success rates vary with different numbers of blocked edges. While most methods achieve a 60%–70% success rate in original environments, the curves sharply decline to around 40% when a single edge is obstructed, with further deterioration observed in the Block-2 and Block-3 scenarios. ScaleVLN outperforms other methods in R2R-UNO owing to its strong generalization ability brought by extensive training data, but it still suffers from a nearly 20% performance reduction in the block-1 set of R2R-UNO. We conclude that current VLN models are overly focused on instruction-following capabilities and lack essential basic navigation functionality to adapt to graph changes.

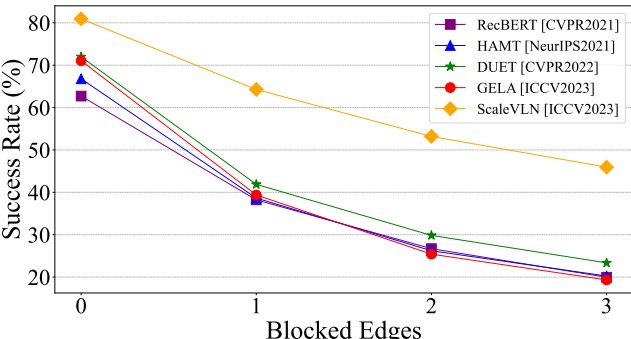

**Figure 4: The large performance drop of current VLN methods in the validation unseen splits of R2R-UNO.**

These findings emphasize the severity of using the perfect instruction assumption in VLN research and the urgency for its resolution.

### 4.2 ObVLN

With R2R-UNO, agents can be trained in obstructed environments to better adapt to instruction-reality mismatches. However, significant gaps exist between original and obstructed environments, such as navigation requirements, reliance on instructions, and trajectory numbers, making direct training on these two types of data compromise the performance in both environments (See Tab. 2).

Therefore, we propose Obstructed VLN (ObVLN), including a novel training strategy and a graph construction mechanism to organize the training of these two environments and help agents deal with obstructions. We advocate a curriculum learning strategy to treat original and obstructed environments as distinct yet complementary tasks to facilitate the training. This strategy leverages the instruction-following skills gained in original environments as a foundation to tackle obstructed settings. Initially, agents are trained in purely unobstructed environments. As training progresses, the sample ratio of obstructed environments $\alpha$ on the training data gradually increases until it reaches a predefined maximum $\alpha_{max}$ at step $c$, defined as follows:

$$\alpha(t) = \min\left(\alpha_{\max}, \frac{t}{c} \cdot \alpha_{\max}\right) \qquad (11)$$

where $t$ is the current training step. All training batches include instances from both settings to ensure smooth optimization.

To facilitate exploration for detours, we introduce a graph construction mechanism for graph-based methods to incorporate "virtual nodes" to represent currently inaccessible nodes due to obstructions. This mechanism consists of three steps, with the pseudo-code in the appendix. Firstly, when encountering an obstacle at node $v_a$, we estimate the location of the obstructed node $v_b$ based on the direction of the obstructed view and a predefined distance $d$ (3 meters in R2R-UNO). This estimation allows us to conceptualize a virtual node $v_b^*$ within the topological graph as a placeholder for $v_b$. Next, as the agent moves, it consistently computes its distances to $v_b^*$ and $v_a$, denoted as $D(v_b^*)$ and $D(v_a)$, respectively. This distance computation is crucial for determining the proximity of the agent to the virtual and obstructed nodes. Finally, if the distances satisfy $D(v_b^*) < \theta$ and $D(v_a) < \theta$ for a given threshold $\theta$, and the current view indicates obstruction, we infer that the agent

**Table 2: Navigation performance of different models on the val seen and unseen splits of R2R and R2R-UNO datasets. "+OE" indicates adding obstructed environments for training.**

| Model | Split | Setting | R2R-UNO-Block-1 | | | | R2R-UNO-Block-2 | | | | R2R-UNO-Block-3 | | | | R2R | | | |
|---|---|---|---|---|---|---|---|---|---|---|---|---|---|---|---|---|---|---|
| | | | TL↓ | NE↓ | SR↑ | SPL↑ | TL↓ | NE↓ | SR↑ | SPL↑ | TL↓ | NE↓ | SR↑ | SPL↑ | TL↓ | NE↓ | SR↑ | SPL↑ |
| HAMT | Val Seen | Basic | 22.50 | 6.01 | 41 | 36 | 28.31 | 7.94 | 28 | 24 | 32.29 | 9.16 | 21 | 19 | 12.53 | 2.68 | **76** | **72** |
| | | +OE | 18.86 | 4.00 | 64 | 59 | 21.08 | 4.86 | 59 | 54 | 23.09 | 5.42 | 55 | 51 | 13.77 | 2.89 | 70 | 66 |
| | | +ObVLN | 18.90 | **3.70** | **66** | **60** | 21.13 | **4.46** | **61** | **56** | 23.02 | **5.02** | **57** | **53** | 12.13 | **2.61** | 75 | **72** |
| | Val Unseen | Basic | 23.05 | 6.84 | 34 | 30 | 28.03 | 8.71 | 22 | 19 | 31.25 | 10.08 | 16 | 14 | 15.21 | 3.67 | 65 | 59 |
| | | +OE | 23.85 | 5.47 | 49 | 42 | 27.43 | 6.70 | 42 | 36 | 29.98 | 7.80 | 35 | 31 | 14.92 | 3.73 | 64 | 58 |
| | | +ObVLN | 23.83 | **5.44** | **51** | **43** | 27.53 | **6.62** | **43** | **37** | 30.70 | **7.65** | **38** | **33** | 14.27 | **3.47** | **67** | **61** |
| DUET | Val Seen | Basic | 18.03 | 6.02 | 50 | 44 | 20.90 | 8.08 | 34 | 30 | 22.95 | 9.47 | 25 | 22 | 14.30 | **2.19** | **80** | **75** |
| | | +OE | 18.66 | 2.87 | 75 | 68 | 19.54 | 2.95 | 74 | 69 | 20.63 | 3.01 | 73 | 69 | 20.19 | 3.13 | 74 | 63 |
| | | +ObVLN | 18.39 | **2.48** | **77** | **71** | 19.25 | **2.59** | **75** | **71** | 20.31 | **2.66** | **74** | **70** | 14.40 | 2.32 | **80** | 72 |
| | Val Unseen | Basic | 19.00 | 6.45 | 44 | 36 | 21.00 | 8.23 | 31 | 25 | 22.15 | 9.51 | 23 | 20 | 16.74 | **3.15** | 72 | **60** |
| | | +OE | 25.04 | 4.01 | 65 | 51 | 25.64 | 4.32 | 63 | 52 | 26.78 | 4.73 | 60 | 51 | 25.02 | 4.02 | 65 | 49 |
| | | +ObVLN | 25.13 | **3.54** | **67** | **53** | 25.44 | **3.78** | **65** | **54** | 26.12 | **4.07** | **63** | **54** | 17.12 | 3.42 | **72** | 57 |

has arrived or moved close to $v_b$. So we connect the virtual node $v_b^*$ and current node $v_c$ with a zero-weight edge, indicating their alignment. This mechanism helps agents to seek alternative routes more purposefully and increase exploration efficiency.

## 5 EXPERIMENTS

### 5.1 Datasets and Evaluation Metrics

We mainly evaluate our methods on the widely used VLN benchmark R2R [5] with step-by-step instructions and our proposed R2R-UNO to focus on the challenge of instruction-reality mismatches. We present results on the goal-oriented benchmark REVERIE [52] without such mismatches in the appendix. R2R is built on the Matterport3D dataset [10], including 10,800 panoramic views from 90 building-scale scenes. Each R2R path has three or four natural language instructions from human annotators. R2R has four splits: train, validation seen (val seen), validation unseen (val unseen), and test unseen. We utilize the train split for training and the val seen and val unseen splits for evaluation to align with R2R-UNO.

For evaluation, we follow previous works [2, 72] to use four primary metrics in VLN: (1) Trajectory Length (**TL**): the total navigation length in meters; (2) Navigation Error (**NE**): the distance between the stop location and the target; (3) Success Rate (**SR**): the ratio of agents stopping within 3 meters of the target; (4) Success rate weighted by Path Length (**SPL**) [4]: SR normalized by the ratio between the length of the shortest path and the predicted path. Metrics related to instruction fidelity like CLS [27] or nDTW [26] are not included due to the modified trajectory with obstacles.

### 5.2 Implementation Details

For R2R-UNO creation, we use the stable-diffusion-v1.5-inpainting model for object insertion and the CLIP ViT-L/14 to evaluate text-image pairs. We use brute force and K-Nearest Neighbors matching to align the SIFT features of two adjacent views. For methods in Fig. 4, we use their best models on the validation unseen split and extract the features of the obstructed environments according to their settings. We employ HAMT [11] and DUET [12] for navigation training and follow the implementation details in their official repositories. The obstructed environments are only used in the

fine-tuning stage. The maximum sample ratio $\alpha_{max}$ is set to 0.5 with the increasing step $c$ as 20,000. We increase the maximum action length to 30 for longer ground truth paths. PREVALENT [22] is used as the augmented data to stabilize the training. We utilize the AdamW optimizer [44] and select the best model based on the average SPL in the val unseen splits of R2R and R2R-UNO. All models are fine-tuned for 100K iterations with a learning rate of 1e-5 and a batch size of 8 on a single NVIDIA A6000 GPU.

### 5.3 Main Results

We first demonstrate that our proposed obstructed environments and ObVLN can facilitate agent adaptation to instruction-reality mismatches. Therefore, we apply HAMT and DUET to three distinct training settings: (1) **Basic**: training with R2R; (2) **+OE**: incorporating data from both the R2R and R2R-UNO; (3) **+ObVLN**: using the ObVLN method on R2R and R2R-UNO. Note that both settings 2 and 3 include all three sets of R2R-UNO. Tab. 2 presents the navigation performance of different models on the val seen and unseen splits of R2R and R2R-UNO. The results show that models trained with two types of data significantly outperform those trained only on R2R in navigating obstructed scenes within the R2R-UNO dataset. However, this improvement is accompanied by a noticeable degradation in R2R performance, like the 6% and 7% SR drop of DUET in the seen and unseen splits, respectively. This phenomenon can be explained by the considerable TL increase in R2R, suggesting that agents are over-optimized for obstructed environments and tend to take detours even without obstructions. Our ObVLN method effectively addresses this issue. For HAMT, ObVLN surpasses the basic setting in R2R, achieving the best results across all four sets of R2R and R2R-UNO. For DUET, the topological map and global action space lead agents to return to the node with the blocked edge which aligns best with the instruction. This results in a slight loss of efficiency in terms of SPL in R2R for ObVLN. However, it still achieves a comparable SR to the basic setting and significantly outperforms setting 2 in R2R. Additionally, it achieves state-of-the-art results in all three sets of R2R-UNO. The advantage of HAMT with ObVLN is less significant because HAMT is not a map-based approach, making our graph construction mechanism inapplicable.

**Table 3: Ablation study on the Object insertion Module (OM) and the Filtering Module (FM) for the val unseen splits.**

| Model | OM | FM | R2R | | R2R-UNO | |
|-------|----|----|-----|------|---------|------|
| | | | SR↑ | SPL↑ | SR↑ | SPL↑ |
| HAMT | × | × | 66.2 | 60.4 | 46.7 | 43.0 |
| | ✓ | × | 65.3 | 59.9 | 48.9 | 44.2 |
| | ✓ | ✓ | **67.1** | **60.8** | **51.7** | **45.6** |
| DUET | × | × | 69.1 | 57.1 | 60.5 | 51.1 |
| | ✓ | × | 71.4 | 58.0 | 64.8 | 53.5 |
| | ✓ | ✓ | **72.3** | **58.3** | **68.5** | **54.9** |

**Table 4: Ablation study of different training strategies on the val unseen splits of R2R and the Block-1 set of R2R-UNO.**

| Sample | $c$ | R2R | | R2R-UNO | |
|--------|-----|-----|------|---------|------|
| | | SR↑ | SPL↑ | SR↑ | SPL↑ |
| Path-wise | - | 66.8 | 51.2 | 65.7 | 52.3 |
| Task-wise | - | 70.9 | 58.1 | 67.0 | 54.5 |
| Instruction-wise | - | 69.4 | 54.9 | 67.7 | 54.3 |
| Ours | 10K | 71.5 | **59.1** | 66.4 | **56.1** |
| | 20K | **72.3** | 58.3 | **68.5** | 54.9 |
| | 30K | 71.6 | 58.0 | 67.0 | 55.4 |

While DUET outperforms HAMT by a 7% SR increment on R2R, this superiority becomes much more significant in obstructed environments, in which DUET achieves around 25% SR lead in the most challenging Block-3 set. We attribute this to the topological map design in DUET, which significantly enhances exploration efficiency and serves as a critical component in finding detours. This advantage is consistent with the large lead of DUET on exploration-intensive, object-oriented datasets like REVERIE [52].

## 5.4 Ablation Study

In this section, we conduct an ablation study on several important components to show their effectiveness.

*5.4.1 Two Modules for R2R-UNO.* We first evaluate the impact of the Object insertion Module (OM) and the Filtering Module (FM) by generating different Block-1 sets for training and evaluation. Without these two modules, the obstructed environments only contain graph changes without visual modifications. Tab. 3 presents the performance of HAMT and DUET on the val unseen splits using different modules. The R2R performance is consistent across all experiments due to ObVLN. For obstructed environments, using the object insertion module to modify the panoramic views can enhance agent navigation by providing critical visual feedback on graph changes, evidenced by performance gains. Moreover, including the filtering module further improves the inpainting quality, leading to more successful obstruction generations and the best performance.

*5.4.2 Sampling Strategy.* We explore four sampling strategies for DUET to utilize both original and obstructed environments: (1) Path-wise, where a path is randomly selected from the combined

**Table 5: Ablation study of different graph construction methods on the val unseen splits of R2R-UNO.**

| Graph | Block-1 | | Block-2 | | Block-3 | |
|-------|---------|------|---------|------|---------|------|
| | SR↑ | SPL↑ | SR↑ | SPL↑ | SR↑ | SPL↑ |
| Vanilla | 63.9 | 49.0 | 60.0 | 48.0 | 56.3 | 46.8 |
| Ours | 67.4 | 53.0 | 65.4 | 54.1 | 63.1 | 54.0 |
| Oracle | **69.8** | **55.0** | **68.0** | **57.1** | **66.2** | **57.0** |

**Table 6: Ablation study of different obstructions on the val unseen splits of R2R-UNO.**

| Split | Version | R2R | | R2R-UNO | |
|-------|---------|-----|------|---------|------|
| | | SR↑ | SPL↑ | SR↑ | SPL↑ |
| Val Seen | 1 | 78.06 | 71.27 | 77.95 | **71.78** |
| | 2 | 78.45 | 70.23 | **78.44** | 70.57 |
| | 3 | 79.14 | 72.68 | 77.56 | 70.05 |
| | 4 | 78.94 | 70.37 | 76.30 | 70.56 |
| | 5 | **79.72** | **73.11** | 77.34 | 69.71 |
| | $\sigma^2$ | 0.33 | 1.38 | 0.51 | 0.49 |
| Val Unseen | 1 | **72.37** | 57.97 | 68.37 | 54.96 |
| | 2 | 71.95 | 57.69 | 68.25 | 53.97 |
| | 3 | 71.52 | **58.60** | **68.66** | 54.52 |
| | 4 | 72.29 | 57.38 | 67.98 | **56.21** |
| | 5 | 71.82 | 58.50 | 67.93 | 53.24 |
| | $\sigma^2$ | 0.10 | 0.22 | 0.07 | 0.99 |

pool of R2R and R2R-UNO paths; (2) Task-wise, where data is sampled from original and obstructed environments in a Bernoulli mixture distribution with probabilities $1 - \alpha_t$ and $\alpha_t$, respectively; (3) Instruction-wise, where for each instruction, one path is randomly chosen from all possible paths. (4) Curriculum sample (Ours), which gradually increases the sampling ratio $\alpha$ up to $\alpha_{max}$ by step $c$. Tab. 4 shows their performances on the the val unseen splits of R2R and R2R-UNO. We set $\alpha_t = 0.5$ to match $\alpha_{max}$ and provide experiments on different $\alpha_t$ and $\alpha_{max}$ in the appendix. Among all strategies, our curriculum sampling achieves dual superiority, achieving the best performance in both R2R and R2R-UNO. Other methods experience performance drops on R2R due to the over-optimization problem. Path-wise sampling obtains the worst performance due to path imbalance, with more training on paths with many redundant edges while ignoring those with none or less.

*5.4.3 Graph Construction Mechanism.* We compare our graph construction mechanism with two baselines: the Vanilla setting that overlooks the obstruction in the graph and an idealized Oracle setting that receives the ground truth information about nodes occluded by obstructions as a performance ceiling. Tab. 5 shows the performance of DUET with different graph construction methods on the val unseen splits of R2R-UNO. Our approach, incorporating virtual nodes into the graph, significantly outperforms the vanilla setting on all three sets, especially in more challenging scenarios with three obstructed edges. As expected, the Oracle graph achieves the highest performance due to its access to accurate, unobstructed

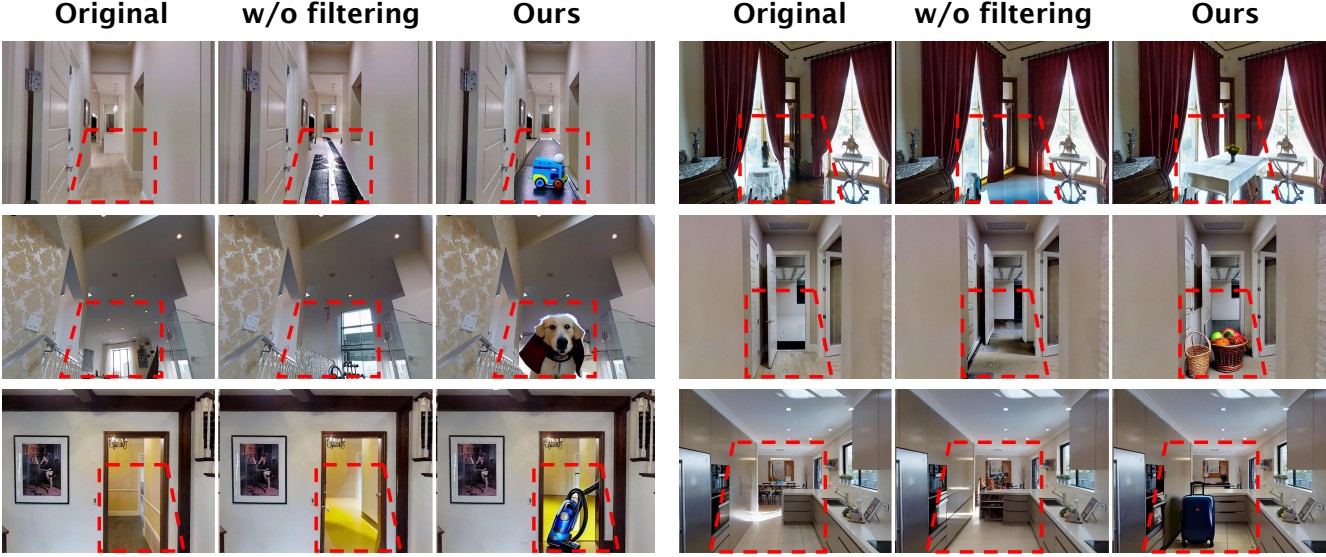

**Figure 5: Qualitative analysis of inpainting results. Left: Original Matterport3D views. Middle: Results without filtering module. Right: R2R-UNO results. The red dash line denotes the mask contour.**

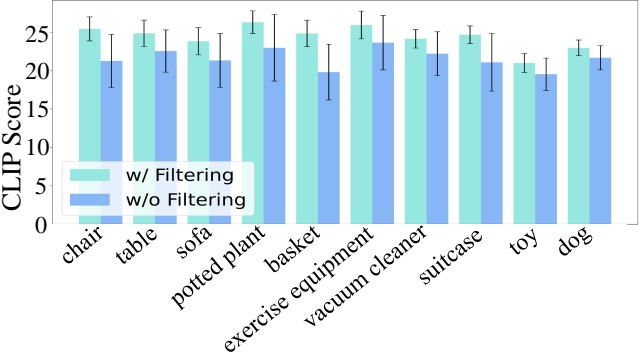

**Figure 6: The comparison of the CLIP score for each category when w/ and w/o the filtering module.**

node information. These findings emphasize the necessity of equipping agents with obstruction-aware capabilities and the benefits of integrating obstructions into the topological map.

*5.4.4 Different obstructions.* To evaluate the generalization ability of our agents to various obstructions, we generate five distinct sets of obstructions for each modified node in R2R-UNO using random objects and seeds through the proposed two modules. We use them to replace the obstructions in R2R-UNO, making five new versions of R2R-UNO that share the same modified graphs and differ only in the obstructions, denoted as R2R-UNO-$i$ for $i \in [1, 2, 3, 4, 5]$. We train our ObVLN model in each version and evaluate the agents across the val unseen splits of all five versions. Due to the low variance in results ($\pm 0.1$), we present the average performance for all versions. As shown in Tab. 6, our agents achieve consistent navigation performance across different datasets, regardless of their training environments. This consistency suggests that our agents can effectively generalize across different obstructions, which is expected since the obstructions are only used for visual feedback and

are irrelevant to subsequent actions. These results further support our claims for addressing the multi-view inconsistency in Sec. 3.2.2.

## 5.5 Qualitative Analysis

We present some obstructed environments from R2R-UNO in Fig. 5 comparing with corresponding original views and those generated without the filtering module. Our method successfully integrates various objects into specific locations within the original views, creating realistic and contextually harmonious obstructions. In contrast, results without the filtering module often fail to include the objects, demonstrating the critical role of this module in enhancing the inpainting reliability. Additionally, we evaluate the generation quality using the CLIP [54] score and present the scores for each category in Fig. 6. The filtering module consistently improves and stabilizes the generation quality, achieving higher scores and lower variance across all categories, aligned with the examples.

## 6 CONCLUSION

This work introduces obstructed environments into VLN to address the prevalent issue of instruction-reality mismatches in real-world navigation. We present R2R-UNO, the first VLN dataset to incorporate such mismatches by integrating diverse obstructions into R2R at both the graph and visual levels through a novel object insertion and filtering module. Using R2R-UNO, we demonstrate that current VLN methods struggle in obstructed settings and further propose the ObVLN method to help agents effectively adapt to obstructed environments. Through these innovations, our agents achieve state-of-the-art results in R2R-UNO while maintaining robust performance in R2R. We believe addressing the perfect instruction assumption is crucial for the practical application of VLN agents and for assessing their adaptive capabilities to navigate beyond only following instructions. Future work should improve the agent performance in both settings and extend this work to continuous 3D environments.

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
