# OpenReview forum: "Navigating Beyond Instructions: Vision-and-Language Navigation in Obstructed Environments"
_acmmm.org/ACMMM/2024/Conference — MM2024 Oral_

### Official Review · Reviewer_vNgx · 2024-05-22

**Rating:** 3
**Confidence:** 2

**Summary:**

The paper introduces the R2R with Unexpected Obstructions (R2R-UNO) dataset and task to address the issue of instruction-reality mismatches in Vision-and-Language Navigation (VLN).  The authors propose the ObVLN (Obstructed VLN) method which includes a curriculum training strategy to organize training on both original and obstructed environments, and a virtual graph construction mechanism to help agents explore efficiently when facing obstructions.

**Strengths:**

(1)The paper is well-structured and clearly written. The motivation, problem formulation, proposed solutions, and experimental results are presented in a logical and easy-to-follow manner.
(2)The proposed R2R-UNO dataset and task are the first to explicitly incorporate such instruction-reality mismatches into VLN benchmarks.
(3)The paper provides comprehensive experiments to validate the significance of the proposed dataset and method.

**Limitations:**

(1)The experiments are limited to simulated environments.  While this is common in VLN research, real-world validation is crucial for demonstrating practical applicability. To fully demonstrate the practicality of the proposed method, it would be beneficial to include real-world experiments or discuss the challenges and potential solutions for deploying ObVLN agents in physical environments.
(2)The object insertion and filtering modules, while innovative, are complex and computationally expensive. This complexity may limit practical implementation due to the reliance on high computational resources and specific inpainting models, which could be a barrier for broader adoption.
(3)The paper mentions potential issues with multi-view inconsistency due to the use of 2D inpainting methods but does not thoroughly discuss the impact this may have on agent performance or possible solutions.

**Suitability:**

3

---

### Official Review · Reviewer_Z2zq · 2024-05-23

**Rating:** 4
**Confidence:** 4

**Summary:**

They introduce R2R-UNO, which contains various types and numbers of path obstructions to generate instruction-reality mismatches for VLN research. To  help agents effectively adapt to obstructed environments, They propose a novel method called ObVLN (Obstructed VLN), which includes a curriculum training strategy and virtual graph construction.

**Strengths:**

1. The R2R-UNO which provided a foundation for subsequent research，is meaningful. They integrate obstructions into existing discrete VLN environments to block the path described by the instruction, resulting in an instruction-reality mismatch.

2. The result of ObVLN method is significant.

**Limitations:**

1. The R2R dataset is based on step-by-step instructions, which differs from practical applications. I would prefer to see their results on the REVERIE dataset. This dataset uses high-level instructions, which is more realistic.

2. The baseline they use is relatively outdated, DUET is already the method for 2021. Currently, many methods[a,b,c] use semantic maps, large models, etc. for navigation, and I believe these methods can solve the problem of environmental changes.
[a] BEVBert: Multimodal Map Pre-training for Language-guided Navigation
[b] March in Chat: Interactive Prompting for Remote Embodied Referring Expression
[c] GridMM: Grid Memory Map for Vision-and-Language Navigation

**Suitability:**

3

---

### Official Review · Reviewer_KtHJ · 2024-05-25

**Rating:** 6
**Confidence:** 4

**Summary:**

This work shows the discrepancy between instructions and reality in real-world navigation. To solve this problem, this work proposed a novel method called ObVLN, including a curriculum training strategy and virtual graph construction for agent to adapt to obstructed environments.

**Strengths:**

Very novel: The work focuses on the instruction-reality mismatches in VLN, which is a important topic in real application and didn't get attention before.

Well-organized: The work is well-organized and easy to follow. It first introduced the R2R-UNO dataset for exploring instruction-reality mismatch. After that, it introduced the proposed ObVLN structure and make performance comparison between proposed method and current VLN methods in obstructed environment.

Convincing. The work made a throughout analysis on performance comparison results and gave explainations for phenomenon. More experiments and explainations are provided in supplementary materials.

**Limitations:**

No obvious limitations in this work.

**Suitability:**

3

---

### Meta-Review · Area_Chair_coPa · 2024-07-02

**Recommendation:** Accept (Oral)
**Confidence:** 4

**Metareview:**

The work addresses instruction-reality mismatches in VLN, an important topic in real-world applications that has previously been overlooked. The reviewers agree on the paper's technical novelty and convincing results. Some concerns were raised in regard to additional baselines, multi view consistency, real world application etc. and most of them were addressed in the author response. Consequently, I am recommending acceptance of this paper.